# miR449 Protects Airway Regeneration by Controlling AURKA/HDAC6-Mediated Ciliary Disassembly

**DOI:** 10.3390/ijms23147749

**Published:** 2022-07-13

**Authors:** Merit Wildung, Christian Herr, Dietmar Riedel, Cornelia Wiedwald, Alena Moiseenko, Fidel Ramírez, Hataitip Tasena, Maren Heimerl, Mihai Alevra, Naira Movsisyan, Maike Schuldt, Larisa Volceanov-Hahn, Sharen Provoost, Tabea Nöthe-Menchen, Diana Urrego, Bernard Freytag, Julia Wallmeier, Christoph Beisswenger, Robert Bals, Maarten van den Berge, Wim Timens, Pieter S. Hiemstra, Corry-Anke Brandsma, Tania Maes, Stefan Andreas, Irene H. Heijink, Luis A. Pardo, Muriel Lizé

**Affiliations:** 1Molecular & Experimental Pneumology Group, Clinic for Cardiology and Pneumology, University Medical Center Goettingen, 37075 Gottingen, Germany; meritwildung@web.de (M.W.); cornelia.wiedwald@biontech.de (C.W.); heimerl.maren@mh-hannover.de (M.H.); lvolcea@gwdg.de (L.V.-H.); stefan.andreas@med.uni-goettingen.de (S.A.); 2Molecular Oncology, University Medical Center Goettingen, 37077 Goettingen, Germany; maikeschuldt@web.de (M.S.); bernard.freytag@stud.uni-goettingen.de (B.F.); 3Department of Internal Medicine V, Saarland University, 66421 Homburg, Germany; christian.herr@uks.eu (C.H.); christoph.beisswenger@uks.eu (C.B.); m5.sekr@uks.eu (R.B.); 4Laboratory for Electron Microscopy, Max Planck Institute for Multidisciplinary Sciences, 37075 Goettingen, Germany; driedel@mpinat.mpg.de; 5Immunology & Respiratory Department, Boehringer Ingelheim Pharma GmbH, 88400 Biberach an der Riss, Germany; alena.moiseenko@boehringer-ingelheim.com; 6Global Computational Biology and Digital Sciences Department, Boehringer Ingelheim Pharma GmbH, 88400 Biberach an der Riss, Germany; fidel.ramirez@boehringer-ingelheim.com; 7Department of Pathology and Medical Biology, University Medical Center Groningen, University of Groningen, 9712 Groningen, The Netherlands; h.tasena@nus.edu.sg (H.T.); w.timens@umcg.nl (W.T.); c.a.brandsma@umcg.nl (C.-A.B.); h.i.heijink@umcg.nl (I.H.H.); 8Groningen Research Institute for Asthma and COPD, University Medical Center Groningen, University of Groningen, 9712 Groningen, The Netherlands; m.van.den.berge@umcg.nl; 9Institute of Neuro- and Sensory Physiology, Goettingen University, 37073 Goettingen, Germany; malevra@gwdg.de; 10Oncophysiology Group, Max Planck Institute for Multidisciplinary Sciences, 37075 Goettingen, Germany; nairamovsisian@gmail.com (N.M.); urrego.dianae@gmail.com (D.U.); pardo@mpinat.mpg.de (L.A.P.); 11Laboratory for Translational Research in Obstructive Pulmonary Diseases, Department of Respiratory Medicine, Ghent University Hospital, 9000 Ghent, Belgium; sharenprovoost@hotmail.com (S.P.); tania.maes@ugent.be (T.M.); 12Department of General Pediatrics, University Hospital Muenster, 48149 Muenster, Germany; tabea.noethe-menchen@ukmuenster.de (T.N.-M.); julia.wallmeier@ukmuenster.de (J.W.); 13Department of Pulmonology, University Medical Center Groningen, University of Groningen, 9712 Groningen, The Netherlands; 14Department of Pulmonology, Leiden University Medical Centre, 2333 Leiden, The Netherlands; p.s.hiemstra@lumc.nl

**Keywords:** COPD, miR449, ciliogenesis, airway, lung, *CDC20B*, microRNA, *miR449a*

## Abstract

Airway mucociliary regeneration and function are key players for airway defense and are impaired in chronic obstructive pulmonary disease (COPD). Using transcriptome analysis in COPD-derived bronchial biopsies, we observed a positive correlation between cilia-related genes and microRNA-449 (*miR449)*. In vitro, *miR449* was strongly increased during airway epithelial mucociliary differentiation. In vivo, *miR449* was upregulated during recovery from chemical or infective insults. *miR0449*^−/−^ mice (both alleles are deleted) showed impaired ciliated epithelial regeneration after naphthalene and *Haemophilus influenzae* exposure, accompanied by more intense inflammation and emphysematous manifestations of COPD. The latter occurred spontaneously in aged *miR449^−/−^* mice. We identified Aurora kinase A and its effector target HDAC6 as key mediators in *miR449*-regulated ciliary homeostasis and epithelial regeneration. Aurora kinase A is downregulated upon *miR449* overexpression in vitro and upregulated in *miR449^−/−^* mouse lungs. Accordingly, imaging studies showed profoundly altered cilia length and morphology accompanied by reduced mucociliary clearance. Pharmacological inhibition of HDAC6 rescued cilia length and coverage in *miR449^−/−^* cells, consistent with its tubulin-deacetylating function. Altogether, our study establishes a link between *miR449*, ciliary dysfunction, and COPD pathogenesis.

## 1. Introduction

Cilia are highly conserved, microtubule-based surface organelles with important mechanical and sensory functions. Almost all mammalian cells build up a single, immotile primary cilium, while specialized epithelial cells lining the airways, reproductive ducts, and brain ventricles form multiple motile cilia per cell that promote directional fluid flow and particle transport along the multiciliated surfaces by coordinated beating. Each cilium on a multiciliated cell (MCC) is nucleated by a basal body that is derived from a centriole; hence, cell cycle exit is a prerequisite for cilia formation in most of the cells [1]. However, upon injury, cilia are disassembled to permit cell proliferation and epithelial repopulation prior to differentiation [2]. Motile cilia are composed of a characteristic axonemal 9 + 2 microtubule architecture that polymerizes from α- and β-tubulin dimer. Another important class of components within the complex ciliary nanomachine are the axonemal dyneins, generating the ciliary beat [1].

In the respiratory system vigorous beating of multiple cilia transports inhaled pollutants and pathogens out of the airways, thereby acting as a primary innate defense mechanism. Functional defects in airway MCCs impair mucociliary clearance and are associated with human pathologies, e.g., primary ciliary dyskinesia (PCD) [3]. As well as MCCs, the pseudostratified epithelium of the conducting airways is lined by club, goblet, and basal cells. All these epithelial cell types are repeatedly subjected to injury by inhaled environmental toxicants and pathogens as well as systemically administered xenobiotics. Hence, rapid and balanced regeneration of the epithelial surfaces is essential to maintain respiratory function [4] and healthy lung aging [5].

The events that trigger multiciliogenesis and lead to its dysfunction in respiratory disease depend on a hierarchical network of transcriptional and post-transcriptional mRNA regulators [6]. Such post-transcriptional regulators include microRNAs (miRNAs), a class of small regulatory noncoding RNAs. miRNAs exert their function by binding to the three prime untranslated regions (3′UTR) of their target mRNAs, thereby causing mRNA degradation or translational repression [7]. In mammalian cells, in most cases, miRNA bind their target with imperfect complementarity. One miRNA family involved in the regulation of multiciliogenesis is the *miR34/449* family derived from three genomic loci encoding six homologous miRNAs (*miR34a*, *miR34b,c*, and *miR449a/b/c*) [8,9,10,11,12,13]. The *miR449* cluster is expressed from the second intron of cell division cycle 20B (*CDC20B*) on chromosome 13 D2.2 in mice (5q11.2 in humans) [14]. This locus contains other key regulators of multiciliogenesis including Cyclin O *(**CCNO)* and multiciliate differentiation and DNA synthesis-associated cell cycle protein (*MCIDAS)* [15,16,17]. The importance of this locus in multiciliogenesis is highlighted by the fact that patients harboring mutations in *CCNO* or *MCIDAS* develop congenital mucociliary clearance disorders with the reduced generation of multiple motile cilia [16,18]. Moreover, *miR449* is by far the most strongly induced miRNA during mucociliary differentiation [8,9]. The depletion of all members of the *miR34/449* family in mice demonstrated their essential function in multiciliogenesis, resulting in infertility, severe respiratory dysfunctions caused by defective mucociliary clearance, and early death from respiratory failure [10,11,12,13].

Increasing air pollution [19] and the demographic shift towards an aging population [20] are associated with the increased prevalence of chronic respiratory diseases worldwide, notably COPD. The prevalence, morbidity, and mortality of COPD are high, especially in the elderly, and it is currently the third leading cause of death worldwide. The main risk factor for COPD is exposure to noxious gases, including cigarette smoke (CS), air pollution, and job-related fumes, resulting in an inflammatory response and tissue damage [21]. Subsequently, aberrant repair and regenerative responses lead to mucociliary dysfunction, reduced defense against pathogens, and, thus, repeated bacterial or viral exacerbations [22,23]. These are thought to be responsible for the progressive destruction of the alveoli by protease–antiprotease imbalance, resulting in airflow limitation and the requirement of oxygen supply. Unfortunately, there is no treatment to cure COPD and lung transplantations are not always possible or successful, and, if so, the incidence of bronchiolitis obliterans syndrome is high. Therefore, there is an urgent need to develop an understanding of the pathomechanisms of chronic pulmonary diseases and the key players in healthy lung aging despite ever-growing challenges, e.g., virus-induced pneumonia as observed in the SARS-CoV-2 pandemic among others [21].

Here, we report for the first time a connection between *miR449* and cilia-related processes in COPD patients. By investigating the effects of *miR449* deficiency on bronchial epithelial regeneration upon different damaging challenges and COPD development in mice, we found that *miR449* is a major contributor to mucociliary regeneration. Surprisingly, it does so by regulating the ciliary assembly–disassembly pathway and its key player Aurora kinase A (AURKA), a centrosomal kinase that regulates mitotic entry but also cilia disassembly. Loss of *miR449* increases AURKA levels, reduces cilia and mucociliary clearance, and even triggers spontaneous emphysematous manifestations of COPD in mice under pathogen-free, barrier maintenance. Pharmacological inhibition of AURKA’s effector target HDAC6 rescues the airway cilia of *miR449^−/−^* in organotypic cultures. Thus, *miR449* protects the airway cilia, mucociliary clearance, and healthy lung aging and should be assessed as a potential regenerative booster in airway diseases with ciliary damage, e.g., COPD frequent exacerbators.

## 2. Results

### 2.1. Cilia-Related Genes Positively Correlate with miR449 Expression in COPD Patients

To elucidate whether *miR449* plays a role in cilia-associated processes, we analyzed genome-wide miRNAs and mRNAs of bronchial biopsies from 57 COPD patients included in the GLUCOLD study [24] (Figure 1a and Appendix A, for details of GLUCOLD study, see Method section). We found 1444 genes to be positively correlated with *miR449a* (FDR < 0.05). In order to determine the biological processes in which the 1444 *miR449a*-correlated genes are involved, gene set enrichment analysis (GSEA) [25] was conducted using a list of genes ranked on the basis of the strength of their correlation to *miR449a*. Several gene sets associated with ciliogenesis showed an enrichment among the genes positively correlated to *miR449a*, including the first five processes shown in Figure 1b. As expected from previous data [8,10,11,12], genes positively correlated with other *miR34/449* family members, except for *miR34a*, were also enriched for cilia-associated processes (Appendix A). To identify genes through which *miR449a* regulates motile ciliogenesis, we compared *miR449a*-positively correlated genes (1444) with a published list containing 581 cilia-associated genes [26] and identified 135 overlapping genes related to the structure, maintenance, and function of cilia (Figure 1c). Among the most significant genes was the dynein intermediate chain 1 *(DNAI1*), encoding an outer dynein arm protein involved in cilia beating [27] (Figure 1c). In conclusion, *miR449a* is positively correlated with ciliary genes in COPD patients.

### 2.2. miR449 Is Induced during Airway Epithelial Differentiation and Regeneration upon Bronchial Challenges

The positive correlation between cilia-associated genes and *miR449* in COPD patients is in line with previously published data showing *miR449* expression in multiciliated epithelia [8,9,10]. To verify *miR449* expression during multiciliated epithelia formation, we analyzed its expression by in situ hybridization (ISH) on mouse embryos (embryonic (E) day 18.5) and found that *miR449a* is enriched in multiciliated tissues, perceptibly in the developing airway epithelium (Figure 2a). Previous studies have already revealed that the *miR449a,b,c* cluster is induced far more strongly than the *miR34* members during mucociliary differentiation [8,9,11,12]. We confirmed these data in our current study by culturing human airway epithelial cells collected from healthy donors at the air–liquid interface (ALI) for 23 days, a time point when human ALI cultures are fully differentiated (Figure 2b). Moreover, RNA-sequencing data from ALI cultures of mouse tracheal epithelial cells (MTECs) supported these results [26] (Figure 2c). The onset of *miR449* expression occurs along with the expression of other known players of multiciliogenesis including Geminin coiled-coil domain containing (*Gemc1*), *Mcidas*, and forkhead box J1 (*Foxj1*) [6,17]. Notably, the total read counts of *miR449* were most strongly upregulated compared to the other pro-ciliogenic factors including *miR34b,c* and were at their highest between day 4 and 7 (Figure 2c). These results indicate that *miR449* has a much stronger contribution to mucociliary differentiation compared with its homologues. Therefore, we focused our following analyses on investigating the role of *miR449* in ciliated airway epithelia regeneration. One commonly used airway regeneration mouse model is the exposure to naphthalene, which leads to extensive club cells’ exfoliation [28] and the squamation of ciliated cells accompanied by motile cilia disassembly [2]. Thus, we treated wild type (WT) mice with naphthalene and assessed the lungs at different stages during the airway epithelia regeneration process, which is normally completed after 14 days (Figure 2d). *miR449a* expression levels immediately increased on day 1 (d1) after naphthalene injection and continued to accumulate until d7, where the expression of *miR449a* peaked (Figure 2e).

Air pollutants such as diesel exhaust particles (DEPs), which are a major constituent of traffic-related particulate matter, represent another environmental challenge for the respiratory epithelium. DEPs are known to result in acute airway inflammation and the reduction in ciliated and non-ciliated cells [29], as well as a decrease in cilia beat frequency [30]. Due to the influence of DEPs on ciliated bronchial epithelial regeneration, we examined *miR449* expression in WT mice after DEP treatment (Figure 2f). Indeed, transcript levels of *miR449a* and its host gene *cdc20b* were strongly increased in DEP-exposed WT mice (Figure 2g). In summary, the data reveal an upregulation of *miR449* during mucociliary differentiation and upon bronchial challenges, implying that *miR449* is involved in the regulation of these two processes.

### 2.3. miR449 Deficiency Results in Impaired Ciliation in ALI Cultures

To further investigate the role of *miR449* in ciliogenesis, we utilized the *miR449^−/−^* mouse model, in which only *miR449a*, *b*, and *c* (*miR449* from here on) are selectively depleted, whereas *miR34a* and *miR34b,c* [10] are not (Figure 3a,b). Previous studies showed that the expression of *miR34b* and *c* is increased upon *miR449* depletion and this can compensate for the loss of *miR449* in the testis [11,31]. Therefore, *miR449^−/−^* mice are fertile, while double knockout *miR34b,c*^−/−^; *miR449^−/−^* mice are infertile [10,11,12]. Interestingly, the lungs of *miR449^−/−^* mice, in contrast to the testis, did not show increased expression of *miR34* members (Figure 3b), as previously shown [11]. Hence, *miR449^−/−^* mice provide the possibility to study *miR449*-mediated cilia regeneration and pulmonary function without effects resultant from *miR34* changes. We measured cilia quantity and cilia length in ALI cultures generated from cells of WT and *miR449^−/−^* mice at d6, a timepoint of the highest *miR449* expression in WT cells (Figure 2c). Immunofluorescence (IF) staining for acetylated-*alpha*-tubulin (Ac-α-TUB) and DNAI1 (both axonemal cilia markers) revealed a significant decrease in the number and length of motile cilia in *miR449^−/−^* compared with WT ALI cultures (Figure 3c–f and Appendix A). Thus, *miR449* deficiency impairs the process of multiciliated epithelium generation in vitro.

### 2.4. miR449 Is Essential for Bronchial Epithelial Regeneration

The formation and proper regeneration of the multiciliated airway epithelium is extremely important for normal respiratory function since the airways are continuously exposed to environmental insults. Since *miR449* is upregulated upon environmental stimuli and mucociliary differentiation is impaired in *miR449^−/−^* mouse derived ALIs, we next examined the potential of ciliated bronchial epithelial regeneration in *miR449^−/−^* mice after naphthalene treatment (Figure 2d). Airway cilia coverage was assessed by IF staining and immunoblot for Ac-α-TUB and DNAI1, respectively, at different regeneration stages. Both axonemal cilia markers were significantly decreased after naphthalene injury (d1–d7), and even at the stage of full bronchial epithelium regeneration in WT mice (d14), ciliation in *miR449^−/−^* mice remained drastically reduced (Figure 4a–c). Hematoxylin and eosin staining did not reveal any obvious histological differences between WT and *miR449^−/−^* lungs after naphthalene injury (Appendix A), although *miR449^−/−^* mice experienced higher weight loss in comparison to WT mice (Appendix A). These findings demonstrate that *miR449* is necessary for adequate bronchial epithelial regeneration.

In addition to toxic and pollution particles, the airway epithelium is constantly exposed to pathogenic bacteria. Therefore, in our next experiment we studied whether *miR449^−/−^* mice are more vulnerable to nontypeable *Haemophilus influenzae* (NTHi) (Figure 4d), an opportunistic respiratory bacterium altering cilia beating and associated with chronic airway diseases such as COPD [32]. Motile cilia coverage and expression of cilia markers were indeed significantly reduced in lungs of NTHi-exposed *miR449^−/−^* mice compared to lungs of NTHi-exposed WT mice, as shown by IF for Ac-α-TUB, immunoblot for DNAI1, and gene expression analysis of the outer dynein arm component dynein axonemal heavy chain 5 (*Dnah5*) (Figure 4e–g).

Although airway epithelial turnover is relatively slow, the epithelium needs to be regularly replaced due to constant challenges during the breathing process, and epithelial regeneration is also important during aging. We thus hypothesized that *miR449* depletion could impair airway ciliation in aged mice. We analyzed IF staining for the Ac-α-TUB and DNAI1 of tracheal sections from 6-month-old WT and *miR449^−/−^* mice. We observed a significant reduction in Ac-α-TUB and an almost complete loss of DNAI1 in 6-month-old *miR449^−/−^* mice (Figure 4h,i), despite being barrier maintained under strict pathogen-free conditions including cage content autoclaving and air filtration. Immunoblot analysis for DNAI1 in lung lysates from WT and *miR449^−/−^* mice confirmed the significant reduction in ciliation in 6-month-old mice upon *miR449* depletion (Figure 4j). In contrast, ciliation appeared normal in 10-week-old *miR449^−/−^* mice (Appendix A). Overall, our data suggest a supportive effect of *miR449* in bronchial epithelial repair upon different challenges including aging.

### 2.5. Spontaneous COPD Development, Associated with an Increased Inflammatory Reaction upon Challenge, in miR449^−/−^ Mice

A reduced lung function as a long-term consequence of COPD is caused by an impaired pulmonary tissue and airway regeneration. In this context, we show the positive effect of *miR449* on bronchial regeneration. We therefore tested whether the loss of *miR449* has any effect on lung function. Aged WT and *miR449^−/−^* mice were subjected to invasive lung function while inhaling ambient air (AA). WT mice exposed to CS for 6 months were used as a positive control due to the loss of lung function and resulting tissue destruction [33] (Figure 5a). Pulmonary function was impaired to a similar extent in *miR449^−/−^* mice that were exposed to AA as well as in CS-exposed WT mice, which was indicated by an increase in total lung capacity and reduction in tissue elastance (Figure 5b,c), whereas no such phenotype observation could be made in young *miR449^−/−^* mice (Appendix A). By stereology, we assessed serial histological lung sections by means of the average chord length, thereby confirming the development of spontaneous emphysematous changes in aged *miR449^−/−^* mice, characterized by airspace enlargement (Figure 5d), which represents one hallmark of COPD. Usually, this condition is associated with an increased infiltration of immune cells, i.a., macrophages and neutrophils, which potentially contribute to the imbalance between proteases and antiproteases within the lungs [34]. We therefore counted the inflammatory cells that were present within the bronchoalveolar lavage fluid (BALF). Under specific pathogen-free conditions, it mainly consisted of macrophages. While the number of macrophages was significantly increased in *miR449^−/−^* compared with WT mice, obviously, the number was not increased to a similar extent as in CS-exposed WT mice (Figure 5e). The pulmonary inflammatory response in mice upon *miR449* deficiency was further analyzed in mice treated with NTHi, an administration used to boost inflammation (as described in Figure 4j). NTHi treatment resulted in elevated protein expression of M2A macrophage marker CD206 in the lungs of *miR449^−/−^* compared with WT mice (Figure 5f,g). Accordingly, the transcript expression of the matrix metalloproteinase (Mmp) 12 (*Mmp12;* macrophage elastase) and *Mmp9* (gelatinase), which are associated with COPD and expressed by macrophages [34], was significantly increased in NTHi-treated *miR449^−/−^* mice (Figure 5h). Contrariwise, we observed unchanged levels of antiprotease upon *miR449* depletion, which were measured by the tissue inhibitor of metalloproteinases 1 (TIMP1) protein expression (Figure 5i,j), suggesting no opposing reaction to the observed increase in the protease level. Together, our data strongly suggest that, with an increasing age, *miR449^−/−^* mice exhibit spontaneous manifestations of COPD. These changes are associated with increased pulmonary MMPs and an increased inflammatory response.

### 2.6. Aurora Kinase A Contributes to miR449-Driven Epithelial Regeneration Processes and Ciliary Homeostasis

Next, we aimed to investigate the underlying mechanism of defective mucociliary regeneration in *miR449^−/−^* mice. In order to avoid pathologies, a balance between proliferation and differentiation is indispensable for airway regeneration, which is, in turn, related to ciliary disassembly and re-assembly. The role of *miR449* in these key processes was studied in a simplified in vitro model given the lack of convenient models available to quantify and analyze the assembly and disassembly of motile cilia. For this purpose, mouse embryonic fibroblasts (MEFs) can be used to monitor and subsequently measure the assembly and disassembly of primary cilia. This is achieved by rounds of serum withdrawal and serum addition, causing cell cycle exit and cell cycle re-entry, respectively [35] (Figure 6a). In the following, MEFs of *miR449^−/−^* and WT mice were analyzed with regard to their cilia assembly and disassembly at the respective points in time, achieved by IF staining of cilia components belonging to the basal body (pericentrin) as well as the axoneme (Ac-a-TUB). Surprisingly, no change in the assembly rate could be observed between both genotypes, whereas ciliary disassembly in *miR449^−/−^* MEFs was significantly increased, in turn resulting in a reduced ciliation (Figure 6b,c). Structural similarities between primary and motile cilia imply mutual mechanisms regarding cilia assembly and disassembly [1]; thus, an increase in the primary cilia disassembly rate could serve as an explanation for the reduction in motile cilia observed in *miR449^−/−^* mice.

Given the increase in the cilia disassembly rate of MEFs from *miR449^−/−^ mice*, we aimed to understand whether *miR449* regulates key players of the cilia disassembly pathway [35,36] (Figure 6d). Potential targets of *miR449* were analyzed with the help of published data [10,12,37,38] as well as in silico prediction tools (TargetScan, PITA) [39,40], subsequently identifying a subset of targets including murine and human histone deacetylase 6 (*HDAC6), AURKA,* and potassium voltage-gated channel subfamily H member 1 (*KCNH1*) (Appendix A). Notably, miRNAs that target multiple nodes within one pathway are more likely to provoke major effects on this pathway. The simple reason behind this is that individual miRNA targets are usually only repressed to a small degree [41]. Due to its essential role in cilia disassembly and HDAC6 activation [42,43,44,45], as well as its upregulation in *miR34^−/−^/449*^−/−^ mice [10,12], we mainly focused on the *miR449* target *AURKA* in the following analysis. *miR449* overexpression in human lung carcinoma cells (H1299) was performed to prove AURKA regulation by *miR449*, and, indeed, the protein as well as the mRNA expression of AURKA was strongly reduced upon overexpression of *miR449* (Figure 6e,f). Appropriately, in lung tissue from *miR449^−/−^* mice, aged and naphthalene-treated, AURKA protein levels were elevated when compared with WT controls (Figure 6g,h). Taken together, these findings suggest reduced airway ciliation to be a consequence of increased ciliary disassembly, triggered by *miR449* deficiency, and probably due to an upregulation of the AURKA/HDAC6-driven disassembly process.

### 2.7. Mice Lacking miR449 Develop Ultrastructural Cilia Defects and Impaired Mucociliary Clearance

AURKA upregulation causes increased cilia disassembly, resulting in primary cilia instability due to the faulty ultrastructure [45,46]. Electron microscopy of tracheal sections showed aberrant cilia with detached plasma membranes (named aberrant cilia from here on) in mice lacking *miR449*, while the typical 9 + 2 microtubule arrangement was maintained (Figure 7a,b). Mucociliary clearance disorders can be caused by defects in structural components of motile cilia [3]. The velocity of fluorescently labelled beads was measured over WT and *miR449^−/−^* tracheae to determine mucociliary transport, with only areas with beating cilia investigated. Indeed, in *miR449^−/−^* mice, bead transport was drastically reduced (Figure 7c,d), indicating impaired clearance. Taken together, these findings show that cilia shape and function are harmed when *miR449* is depleted.

### 2.8. In miR449^−/−^ Airway Cultures’ Ciliation Can Be Rescued by HDAC6 Inhibition

We investigated whether airway ciliation may be restored by blocking the AURKA/HDAC6-mediated cilia disassembly axis. To that aim we used the commercially available selective HDAC6 inhibitor tubastatin [47] to treat ALI cells as HDAC6 is a downstream effector of AURKA (Figure 6d) and a predicted *miR449* target (Appendix A).

In comparison to DMSO-treated cells, *miR449^−/−^* cells treated with an HDAC6 inhibitor have more and longer cilia as shown by IF labeling for the axonemal cilia marker Ac-α-TUB (Figure 8a–c). Thus, inhibition of HDAC6 rescues at least some hallmarks of aberrant ciliation in *miR449^−/−^* airway cultures.

In this study we show that *miR449* loss triggers the deficient ciliated regeneration of respiratory epithelia upon environmental insults and aging. This is characterized by reduced airway cilia and faulty particle transport, and accompanied by increased hallmarks of the murine COPD phenotype, i.e., emphysematous and inflammatory changes. Upregulation of AURKA contributes, at least partially, to greater ciliary instability, probably through hyperactivation of its ciliary effector HDAC6, and both the number and the length of respiratory cilia from *miR449^−/−^* cells were restored upon inhibition of HDAC6 activity. Our research uncovers a novel role for *miR449* in cilia maintenance and healthy lung aging and emphasizes the significance of balanced ciliated regeneration for pulmonary health.

## 3. Discussion

*miR449* has been described in ciliogenesis in earlier studies [8,9,10,11,12,13,48], but this is the first report of its connection with cilia maintenance in response to environmental challenges, and its protective effects on airway biology and function in mouse models of chronic lung diseases and exacerbations. Here we show that the loss of *miR449* impairs ciliated epithelial regeneration and mucociliary clearance by increasing AURKA-driven ciliary disassembly (Figure 9). Thus, *miR449^−/−^* mice develop spontaneous COPD accompanied by an exacerbated pulmonary inflammatory response to challenge.

Previous studies on the *miR34/449* family using genetic depletion of all *miR34/449* family members describe a strong ciliary phenotype of PCD accompanied by defective development and early lethality [10,11,12], thus impairing the long-term analysis of the contribution of *miR449* to disease etiology. In this setting, one allele of *miR449* or of *miR34b,c* was sufficient to avoid complete ciliary disruption [10,12], highlighting the strength of one single miRNA allele. Indeed, here we show that the sole loss of *miR449* increases susceptibility to chronic lung diseases by disrupting an important innate lung defense mechanism, i.e., mucociliary clearance and epithelial regeneration, thus increasing damage in response to environmental challenges. As *miR34b,c* are also expressed in the lungs, although to a lesser extent [8,9,10,11,12,49], it would be insightful to challenge *miR34b,c* knockout (KO) lungs in similar settings, which was beyond the scope of the present study.

It has been proposed that Notch signaling plays an important role in the *miR449*-mediated regulation of ciliogenesis [9]. However, since the deregulation of Notch, which lies upstream of MCIDAS/GEMC1, would influence the ciliated cell’s fate [17], it is not likely to be involved in our findings. As determination of the ciliated cell’s fate was not affected by *miR-34/449* depletion [10,12] and Notch was not upregulated in the *miR-34/449* KO nasal respiratory epithelium [12], our recent data suggest the limited contribution of *miR449*-mediated Notch regulation in mouse cells, although it seems to play a role in the ciliated embryonic frog epidermis [9]. Instead, here we identified that *miR449* deficiency decreases airway ciliation by increasing the ciliary disassembly rate, probably through the upregulation of AURKA and potential overactivation of its downstream effector HDAC6, thereby contributing to the development of manifestations of COPD. Interestingly, significant upregulation of AURKA in response to *miR-34/449* depletion has not only been observed in airways [10,12] but also in fallopian tubes [12], an organ covered by MCCs. Notably, AURKA and its direct target HDAC6 are upregulated in COPD lungs, where, at least, HDAC6 contributes to increased cilia destabilization or “ciliophagy” upon environmental challenge, e.g., cigarette smoking [44,50].

Interestingly, increased AURKA levels generated aberrant primary cilia with ultrastructure defects [45,46] similar to those observed in motile cilia in our study. This, together with HDAC6-mediated motile cilia destabilization in response to smoke [44,50], supports the notion that motile and primary cilia share conserved generation/resorption pathways, and both use the AURKA/HDAC6 axis. As primary cilia biology research has generated tremendous data on this topic [35,51], looking into candidates stabilizing primary cilia might be an interesting path to new targets in respiratory disease if expressed in the airways. One such candidate is the predicted *miR449* target *KCNH1*, which encodes a potassium voltage-gated channel (Kv.10.1) and promotes primary cilia disassembly [36]. Notably, a single nucleotide polymorphism (SNP) mutation in *KCNH1* was associated with COPD susceptibility [52], although the effect of this SNP on Kv10.1 channel function is currently unknown. One limitation of studying miRNA-based regulation is that we cannot exclude the contribution of further targets to the observed mechanisms, especially as we used a systemic KO mouse model. Nevertheless, *miR449* expression was only found in ciliated epithelia [8,9,10,12], suggesting its main functions there, but not excluding communication with other cells, e.g., by exosomes [53]. In this study we showed that the transcript levels of the *miR449* host gene *Cdc20b* and *miR449a* itself were strongly upregulated in DEP-treated WT mice, in the model of particle-induced airway disease. Interestingly, a recent study demonstrated that *mir449a* had an antifibrotic effect in silica-induced lung fibrosis via the activation of autophagic activity in vitro [54]. Utilizing single cell data from Adams et al. [55], we could show that *CDC20B* is expressed preferentially in ciliated cells (Figure 10a). The observed reduction in *CDC20B* expression in COPD-derived cells suggests the concurrent reduction in *miR449* (Figure 10b,c). Interestingly, this dataset shows an increase in *AURKA* and *HDAC6* expression in COPD-derived ciliated cells, mirroring our molecular data. Additionally, new reports have shown that miR449 target HDAC6 is involved in inflammasome assembly in macrophages, suggesting anti-inflammatory effects of *HDAC6* reduction, and many groups incl. pharmaceutical companies have started drug discovery programs to inhibit HDAC6 functions. Importantly, it is very rare to see such a strong differential expression of a miRNA target in a miRNA KO mouse, as observed here for AURKA, due to the mainly fine-tuning functions of miRNAs and their putative compensatory mechanisms such as the proteasomal degradation of excess protein [56]. Therefore, we assume that increased AURKA levels and/or HDAC6 activity play a pivotal role in the development of the COPD-like disorders in the *miR449^−/−^* mouse model. The data presented in this study show a supportive effect of *miR449* on bronchial epithelial regeneration during the aging process and upon various challenges.

Respiratory motile cilia are of utmost importance for lung health, as underlined by the fact that any single non-synonymous disruptive mutation in any core motile cilia protein generates lung diseases ranging from mucociliary clearance disorder with increased susceptibility to recurrent respiratory infections to severe PCD [3]. However, the link between mucociliary clearance and COPD has not been clearly made. The progression of COPD is driven in most cases by recurrent pathogen-driven exacerbations, which usually target the bronchial epithelium at the first site. Immobilization of the respiratory cilia by the ciliostasis of otherwise functional cilia is sufficient to increase the susceptibility to influenza A infection [57], a well-known trigger of COPD exacerbations. Many pathogens have developed ciliostatic factors to abrogate mucociliary clearance and avoid being evacuated from the lungs, e.g., *Pseudomonas* [58,59], *Haemophilus influenzae* [60,61], and fungi [62,63], all typical colonizers of the COPD lung [64].

When pathogens manage to escape mucociliary clearance, they can colonize the lungs. In response to such infections, immune recruitment occurs, and inflammatory signaling is generated including the induction of destructive proteases such as MMP9/12 and neutrophil elastase. Those proteases attack the structural proteins (e.g., elastin) responsible for the recoil properties of the lungs, i.e., elastance and compliance. This suggests that the reduced clearance of pathogens and inflammatory responses associated with airway infections has an impact on the lung function in the long term. Indeed, it is known that exacerbations of COPD, which are often triggered by viral or bacterial infection, are associated with a more rapid decline in lung function. The more often exacerbations happen, the more destruction occurs, and full resolution of the inflammation becomes difficult [21,23,65]. Therefore, hindering pathogen entry by protecting the clearance process might halt COPD progression. Motile cilia are the major factor contributing to mucociliary clearance, and ciliary length [66] and ciliary beat frequency [67] are reduced in COPD. Thus, protecting healthy regeneration of the motile airway cilia, including length and beating ability [47], as well as microbial ciliostasis, might represent new therapeutic approaches to tackle COPD and reduce severe exacerbation rates on top of antiinflammatory drugs. Whether miR449 supplementation may be an option to protect the bronchial barrier is yet unclear, but tool inhibitors addressing aurora kinases and HDAC6 deacetylase activities are already available to study pathway contribution to chronic lung diseases in preclinical models.

## 4. Methods

### 4.1. Correlation of miR449a and mRNAs in COPD Bronchial Biopsies

Bronchial biopsy specimens were collected from 57 COPD patients who were part of the Groningen Leiden Universities and Corticosteroids in Obstructive Lung Disease (GLUCOLD) study [24]. All patients were stable, either current or ex-smokers, and not on corticosteroid therapy. They had irreversible airflow limitation (post-bronchodilator forced expiratory volume in 1 s (FEV1) and FEV1/inspiratory vital capacity < 90% confidence interval of the predicted value) and chronic respiratory symptoms. The local medical ethics committee approved the study, and all patients gave their written informed consent. The study is filled at clinicaltrial.gov under identification number NCT00158847.

mRNA and miRNA profiling were performed at Boston University Microarray Resource Facility using GeneChip^®®^ Whole Transcript Sense Target Labeling Assay and FlashTagTM Biotin HSR Labeling Kit, respectively. Detailed methods on the profiling and processing of microarray data were described previously [68]. Using the matched mRNA and miRNA profiles of the same patient, Pearson correlation coefficient was used to identify genome-wide mRNAs of which expression was correlated with *miR449a* expression. Benjamini–Hochberg procedure was applied for multiple test correction. Significant correlation was defined using false discovery rate adjusted *p*-value (FDR) < 0.05.

To assess the role of *miR449a* in cilia development and function, the list of mRNAs positively correlated with *miR449a* was compared to the previously reported list of cilia-associated genes [26]. Moreover, gene set enrichment analysis (GSEA, v3.0) [25] was performed to identify biological processes (Gene Ontology, v6.1) in which miR449 plays a role. Genome-wide mRNAs were ranked based on their t-statistics representing the strength of their correlation with *miR449a* expression. A *p*-value was calculated after 1000 permutations were performed.

### 4.2. Mice

*miR449^−/−^* mutants had a targeted deletion of *miR449a*, *b*, and *c* and were previously described [10]. WT and *miR449^−/−^* mice were maintained in C57Bl/6 background (n8) at the animal facility of the European Neuroscience Institute Goettingen (Germany) in full compliance with institutional guidelines. The Animal Care Committee of the University Medical Centre Goettingen and the authorities of Lower Saxony approved the study under the number G12/963. WT (C57BL/6) mice used for DEP experiments were obtained from Harlan Laboratories (Horst, The Netherlands) and maintained in the animal facility of the Ghent University. The Animal Ethical Committee of the Faculty of Medicine and Health Sciences of Ghent University approved the in vivo manipulations using DEP. Experiments including the exposure to cigarette smoke (CS) and nontypeable *Haemophilus influenzae* (NTHi) [69,70] were approved by the “Landesamt für Soziales, Gesundheit und Verbraucherschutz” of the State of Saarland following the national guidelines for animal treatment. During the experiment WT and *miR449^−/−^* mice were maintained under specific pathogen-free conditions in the animal facility of the Institute for Experimental Surgery of the Saarland University Homburg.

### 4.3. Cell Culture

H1299 (human non-small cell lung carcinoma) cells were cultured in Dulbecco’s Modified Eagle Medium (DMEM Gibco, Thermo Fisher Scientific, Waltham, MA, USA) supplemented with 10% fetal calf serum (FCS), 2 nM L-glutamine and 10 µg/mL Ciprofloxacin 500. Mouse embryonic fibroblasts (MEFs) were isolated from embryos from WT and *miR449^−/−^* matings, respectively, at embryonic day 13.5 (E13.5) and cultured in DMEM-high glucose (Gibco, Thermo Fisher Scientific) supplemented with 10% heat-inactivated FCS, 100 µg/mL streptomycin, and 100 U/mL penicillin.

### 4.4. Transfection of Human Cells

H1299 cells were reverse transfected with 12 nM (RNA analysis) or 20 nM (protein analysis) synthetic pre-miRNAs (Ambion, Thermo Fisher Scientific) of the *miR34/449* family using Lipofectamine 2000 (Invitrogen, Thermo Fisher Scientific) for transcript and protein analysis, respectively. Scrambled RNA oligonucleotide (negative control #2 (Ctrl)) and naturally occurring pre-*miR-302** were used as controls. Cells were harvested for protein and RNA isolation 48 h (h) after transfection.

### 4.5. Analysis of Primary Cilia Assembly and Disassembly

Freshly isolated WT and *miR449^−/−^* MEFs were plated on fibronectin (Merck, Darmstadt, Germany)-coated coverslips and exposed to serum-free medium (DMEM supplemented only with 100 µg/mL streptomycin and 100 U/mL penicillin) for 60 h to induce primary cilia assembly. Afterwards, serum was reintroduced for 6 h to trigger cilia disassembly. To quantify the percentage of ciliated cells at the assembly and disassembly time point, MEFs were fixed with 2% formaldehyde (Merck) at RT for 8 min and blocked with 10% bovine serum albumin (BSA) (Merck) in phosphate-buffered saline containing 0.1% Triton X-100 (PBS-T) for 1 h 30 min at RT. MEFs were first incubated with the primary antibodies diluted in blocking solution overnight at 4 °C (list of antibodies is provided in Appendix A) and, subsequently, with secondary antibodies for 1 h 30 min at RT (list of antibodies is provided in Appendix A). Nuclei were counterstained with DAPI. Stainings were visualized using a Leica SP2 laser scanning confocal microscope (Leica Microsystems, Wetzler, Germany). Primary cilia were identified by a double staining of pericentrin and Ac-α-TUB. Only cells positive for both were counted as ciliated cells.

### 4.6. Air–Liquid Interface Cultures

MTECs were isolated and cultivated using ALI conditions as described previously [26]. Note that, minor changes regarding the culture and differentiation medium were applied as highlighted below. Briefly, tracheae from WT and *miR449^−/−^* mice (10–16 weeks old) were excised and after pronase digestion (Merck), epithelial cells were seeded onto collagen-coated transwell culture inserts (12 mm diameter, 0.4 µm pores, polyester, Corning, NY, USA). Cells were cultured under submerged conditions in airway epithelial cell growth medium supplemented with growth factors (Promocell, Heidelberg, Germany) 100 U/mL penicillin, and 100 µg/mL streptomycin to allow proliferation. Upon confluence, ALI condition (d0) was generated by removing the apical chamber medium and changing the basolateral medium to the differentiation medium consisting of DMEM/F-12 (Gibco) supplemented with 2% Ultroser G (Pall, Port Washington, NY, USA). For the HDAC6 inhibitor experiment, HDAC6 inhibitor (100 nM, Tubastatin A hydrochloride, Merck, Darmstadt, Germany) or DMSO (control) was added to the basolateral medium of *miR449^−/−^* ALI cultures twice a day from differentiation day (d) 0 to d6. At d6, the entire ALI culture membrane was fixed with 4% paraformaldehyde (PFA) and embedded in paraffin. For one ALI culture, MTECs pooled from three mice were used. Human ALIs were performed in a former study [8], and only RNA was recovered to complete the presented work.

### 4.7. In Situ Hybridization

Tissue cryosections from mouse embryos were fixed with 4% PFA for 10 min, acidified (1.3% Triethanolamine, 0.000065% HCl, 0.25% Acetic Anhydride), and digested with Proteinase K (0.5 μg/mL). After pre-incubation of slides with hybridization buffer (50% Formamide, 25% 20× saline-sodium citrate buffer (SSC) pH 4.5, 1% 0.5 M EDTA, 0.1% Tween-20, 0.1% 3-[(3-cholamidopropyl) dimethylammonio]-1-propanesulfonate (CHAPS), 0.1 mg/mL Heparin, 1% yeast transfer RNA) for 4 h at RT, a dioxigenin-labelled LNA-probe (5 pM) against *miR449a* diluted in hybridization buffer was added for 24 h at 50 °C in a chamber humidified with 50% formamide and 25% 20× SSC buffer. Following two washing steps with pre-warmed 5× SSC buffer for 5 min and 0.2× SSC buffer for 1 h at 50 °C, slides were incubated with buffer B1 (0.1 M TRIS pH 7.5, 0.15 M NaCl) for 10 min at RT and blocked (10% FCS, 0.05% Tween-20, 90% B1 buffer) for 1 h at RT. The alkaline phosphatase-conjugated anti-dioxigenin antibody (Roche Diagnostics, Rotkreuz, Switzerland) diluted in blocking solution (1:2000) was incubated on the slides overnight at 4 °C. After washing with B1 buffer, the slides were incubated with buffer B2 (0.1 M TRIS pH 9.5, 0.1 M NaCl, 50 mM MgCl2, 0.1% Tween-20, 2 mM Levamisole) for 10 min. To visualize the in situ hybridization (ISH) signal, slides were incubated with nitroblue tetrazolium chloride (75 mg/mL)/5-bromo-4-chloro-3-indolyl-phosphate (50 mg/mL) in B2 buffer for up to 3 days and subsequently mounted with fluorescence mounting medium. As a control, slides were treated the same but without addition of LNA probe. Anti-miR449a (ACCAGCUAACAAUACACUGCCA) LNA probe was purchased from Qiagen (Hilden, Germany).

### 4.8. Naphthalene-Induced Injury Model

Naphthalene (Merck) was dissolved in sunflower oil to a concentration of 30 mg/mL in a sterile environment on the day of use. Next, 12–14-week-old male WT and *miR449^−/−^* mice were injected with naphthalene (200 mg/kg) or with oil (vehicle control) intraperitoneally (i.p.) in the morning on day 0.

### 4.9. DEP-Induced Acute Inflammation Model

The DEP instillation was performed as described previously [71,72]. Briefly, DEP (standard reference material (SRM) 2975) was obtained from the National Institute for Standards and Technology (Gaithersburg, MD, USA) and suspended in saline containing 0.05% Tween 80 to a concentration of 2 mg/mL. Next, 6–8-week-old anesthetized mice (i.p. ketamine (70 mg/kg; Ketamine 1000 CEVA, Ceva Sante Animale, Libourne, France)—xylazine (7 mg/kg; Rompun 2%, Bayer, Leverkusen, Germany)) were held vertically and 50 µL saline or DEP solution (100 µg) was pipetted just above their vocal cords. By grasping the tongue, the mice were prevented from swallowing. Mice were instilled at d1, d4, and d7 and sacrificed at d9 by a lethal dose of i.p. pentobarbital (Ceva Sante Animale).

### 4.10. NTHi-Induced Chronic Inflammation Model

A clinical isolate of NTHi was grown on selective chocolate agar with IsoVitaleX at 300 µL per 10 cm plate (Becton Dickinson, Heidelberg, Germany) for 24 h at 37 °C in 5% CO_2_. After harvesting, bacteria were incubated for 24 h in 700 mL brain–heart infusion broth (Becton Dickinson) containing 3.5 mg/mL β-Nicotinamide adenine dinucleotide (NAD, Merck) and 5% Fildes enrichment (Becton Dickinson). The culture was centrifuged at 2500× *g* for 15 min (min) at 4 °C, washed, resuspended in 20 mL PBS, heat inactivated at 70 °C on a mechanical shaker for 45 min, and then sonicated three times for 30 s. The protein concentration was adjusted to 2.5 mg/mL in PBS using the Pierce BCA protein assay (Thermo Fisher Scientific). For bacterial challenge, 4-month-old WT and *miR449^−/−^* mice were placed in a plexiglass box connected to a Pari MASTER^®®^ nebulizer (Pari GmbH, Starnberg, Germany) and exposed to the inactivated bacterial lysate for 40 min 5 times per week for 3 months.

### 4.11. Cigarette Smoke (CS)-Induced Emphysema Model

Four-month-old WT and *miR449^−/−^* mice were exposed to CS (3R4F cigarettes, College of Agriculture, Reference Cigarette Program, University of Kentucky) in a TE-10 smoking machine (Teague Enterprises, Woodland, CA, USA). The concentration of CS was 120 mg/m [3] total suspended particles. Mice were subjected to CS for 5 h per day and 5 days per week over a period of 6 months. The daily smoking protocol consisted of three smoking phases each with 87 min, which were interrupted by room air exposures for 40 min.

### 4.12. Pulmonary Function

Pulmonary function measurement was performed using a FlexiVent system (Scireq Inc., Montreal, QC, Canada). The mice were anesthetized by a mixture of ketamine (Ursotamin, 100 mg/mL, Serumwerk Bernburg, Bernburg, Germany) and xylazine (Rompun, 7 mg/kg, Bayer). The trachea of the anesthetized mouse was exposed and cannulated using the 18 G cannula included with the FlexiVent system. For data acquisition we used the flexiWare 7.1 software (Scireq Inc.).

### 4.13. Bronchoalveolar Lavage

Prior to bronchoalveolar lavage (BAL) the animals were deeply anesthetized with a mixture of ketamine (Ursotamin, 105 mg/mL, Serumwerk Bernburg) and xylazine (Rompun, 7 mg/kg, Bayer). After reaching anesthesia the trachea was exposed and cannulated. The BAL was performed by rinsing the lungs 3 times repeatedly with 1 mL sterile PBS containing protease inhibitors (cOmplete ULTRA Tablets, Mini, Roche Diagnostic). The collected BALF was centrifuged for 5 min at 300× *g*. The number of viable cells was determined by resuspending the cell pellet in 1 ml of sterile PBS and counting on an improved Neubauer chamber after trypan blue staining. For the cellular composition of the BALF a cytospin was stained with DiffQuik (Medion Diagnostic, Gräfelfing, Germany), and at least 200 cells counted and differentiated based on their morphology.

### 4.14. Stereology

The lung was prepared for uniform random sampling for stereological analysis as described elsewhere [73]. Briefly, the lung was fixed by instillation of freshly prepared PBS-buffered 4% formalin under a constant hydrostatic pressure of 20 cm for 20 min. After that, the trachea was ligated to preserve intrapulmonary pressure. For additional fixation, the lung was placed in PBS-buffered 4% formalin for 24 h. The lung volume used as reference for stereological measurements was determined by fluid displacement. The fixed lung was embedded in agar and cut into regular slices prior to paraffin embedding. Stereological analysis was performed on hematoxylin and eosin (H&E)-stained lung sections. Mean chord length (mean linear intercept, L_m_) was calculated using the Visiopharm Integrator System package (Visiopharm, Hoersholm, Denmark) on an Olympus BX51 microscope equipped with an 8-position slide loader (Olympus, Tokyo, Japan).

### 4.15. Histology and Immunostaining

PFA-fixed, paraffin-embedded tissue sections were dehydrated and treated with heat-induced epitope retrieval (sodium citrate, pH 6.0), while cryosections were fixed with 4% PFA for 10 min at room temperature (RT). Following a blocking step with 10% FCS diluted in PBS-T (for cryosections: 1% BSA in PBS-T), primary antibodies diluted in blocking solution were incubated on the samples overnight at 4 °C (list of antibodies is provided in Appendix A). After washing with PBS-T, sections were stained with fluorescently-labelled secondary antibodies for 1 h at RT (list of antibodies is provided in Appendix A). Nuclei were counterstained with 4′, 6-Diamidin-2-phenylindol (DAPI) and samples were mounted with Dako fluorescence mounting medium. For histology, PFA-fixed, paraffin-embedded lung sections were dehydrated and stained with H&E. Images were acquired on a ZEISS AxioScope A1 microscope (ZEISS, Oberkochen, Germany), except tracheae images, which were viewed on a ZEISS confocal LSM 510 or 880 microscope.

### 4.16. Quantification of Cilia Markers

The expression of cilia markers in immunofluorescence images (converted to 8-bit) was quantified using the *ImageJ* software [74]. Briefly, after selecting the region of interest, unspecific signals were removed, and a threshold was set to identify the positive signal. To measure the area of the positive cilia staining the *Analyze Particle* tool was applied. Values of the cilia area were normalized to the length of the respiratory epithelium measured. For length measurements of motile cilia in ALI cultures, the line tool from the *ImageJ* software was used. Cilia length was defined by the area positive for both cilia markers used.

### 4.17. RNA Extraction and Quantitative PCR

RNA from snap-frozen lung tissue and cells (H1299) was isolated by Extrazol (7BioScience, Hartheim, Germany)/Chloroform as recommended by the manufacture. The miRVana kit (Invitrogen, Thermo Fisher Scientific) was used for RNA isolation from human ALI cultures. cDNA from 1 µg RNA was synthesized using the M-MuLV reverse transcriptase (New England Biolabs, Ipswich, MA, USA) and a mixture of random nonameric and Oligo-dT primers (Metabion, Planegg, Germany). For the qPCR reaction, a self-made SYBR Green qPCR Mix (Tris-HCl (75 mM), (NH_4_)_2_SO_4_ (20 mM), Tween-20 (0.01% *v*/*v*), MgCl_2_ (3 mM), Triton X-100 (0.25% *v*/*v*), SYBR Green I (1:40,000), dNTPs (0.2 mM) and Taq-polymerase (20 U/mL)), and 250 nM of each gene-specific primers were used (list of primers is provided in Appendix A). The relative expression of transcripts from each sample was obtained by averaging the cDNA levels of technical triplicates and normalizing it to the housekeeping gene ribosomal phosphoprotein P0 (*Rplp0*, or *36b4*). For miRNA quantification, TaqMan MicroRNA Assay (Applied Biosystems, Thermo Fisher Scientific) for murine and human members of the miR34/449 family was performed according to the manufacturer’s instructions with U6 snRNA as reference gene.

### 4.18. Western Blot

Samples were lysed in RIPA buffer (20 mM Tris-HCl pH 7.5, 150 mM NaCl, 9.5 mM EDTA, 1% Triton X100, 0.1% SDS, 1% sodium deoxycholate) supplemented with urea (2.7 M) and protease inhibitors (Complete Mini EDTA-free, Roche Diagnostic). Note that, for lung samples, lungs were snap-frozen in liquid nitrogen and mechanically homogenized. Equal amounts of protein lysates were resolved by SDS-polyacrylamide gels followed by a transfer to nitrocellulose membranes. Membranes were incubated with primary antibodies overnight at 4 °C (list of antibodies is provided in Appendix A), washed, and stained with horse radish peroxidase-conjugated secondary antibodies for 1 h at RT prior to chemiluminescence detection (list of antibodies is provided in Appendix A). The volume intensity of specific protein bands was quantified using the *Image Lab* software (Biorad, Hercules, CA, USA).

### 4.19. Electron Microscopy

Trachea samples from 7–9-week-old WT and *miR449^−/−^* were fixed by immersion using 2% glutaraldehyde in 0.1 M cacodylate buffer (Science Services, München, Germany) at pH 7.4 overnight at 4 °C. Post-fixation was performed using 1% osmium tetroxide diluted in 0.1 M cacodylate buffer. After pre-embedding staining with 1% uranyl acetate, tissue samples were dehydrated and embedded in Agar 100 (Plano, Wetzlar, Germany). To rule fixation artefacts out, a second embedding technique was performed. For this, samples were placed in aluminum platelets of 150 µm depth containing 1-hexadecen [75]. The platelets were frozen using a Leica EM HPM100 high pressure freezer (Leica Microsystems, Wetzlar, Germany). The vitrified samples were transferred to an Automatic Freeze Substitution Unit (Leica EM AFS2) and substituted at −90 °C in a solution containing anhydrous acetone, and 0.1% tannic acid for 24 h and in anhydrous acetone, 2% OsO_4_, and 0.5% anhydrous glutaraldehyde (EMS Electron Microscopical Science, Hatfield, PA, USA) for additional 8 h. After a further incubation over 20 h at −20 °C, samples were warmed up to +4 °C, and washed with anhydrous acetone. The samples were embedded at RT in Agar 100 at 60 °C for 24 h. Thin trachea sections (100 nm) were examined using a Philips CM 120 BioTwin transmission electron microscope (Philips Inc., Amsterdam, The Netherlands) and images were taken with a TemCam F416 CMOS camera (TVIPS, Gauting, Germany). Longitudinal- and cross-sections from motile cilia were evaluated and grouped in normal and aberrant cilia. Aberrant cilia were defined as cilia with a detachment of the outer membrane. Images were taken and quantified from at least two different tissue locations from each sample.

### 4.20. Mucociliary Transport Assay

Mucociliary transport assay was performed as described in [26]. Briefly, freshly isolated vital tracheae were longitudinally cut using a microtome (VT1200S, Leica Microsystems) and rinsed with Ringer’s solution containing 98 mM NaCl, 2 mM KCl, 1 mM CaCl_2_, 2 mM MgCl_2_, 5 mM glucose, 5 mM sodium pyruvate, 10 mM Hepes (230 mOsm, pH 7.8). Fluorescent microspheres (FluoSpheres^®®^, 0.17 μm, PS-Speck, Invitrogen, Thermo Fisher Scientific) diluted in Ringer’s solution were added onto the multiciliated epithelium of the tracheae. In areas of 280 µm × 280 µm (512 × 512 pixels) the movement of the fluorescent microspheres was recorded using a custom-built confocal line illumination microscope at 61 Hz for 2000 frames. Note that only areas with beating cilia were imaged. For each frame, particles were detected and tracked using functions previously described [26]. Absolute full-trajectory velocities along the focal plane were combined within one measurement. Averaged velocities of independent measurements from different tissue areas and different mice were assigned into the following groups: WT, *miR449^−/−^*, and diffusion. Diffusion measurements were obtained from non-living dried tracheae and pooled from WT and *miR449^−/−^* mice since they were identical. Individual trajectory velocities from all measurements within the respective group were included to obtain particle velocity distributions. All data evaluation was carried out using a custom software written in Matlab (MathWorks, Natick, MA, USA). Age-matched WT and *miR449^−/−^* mice (12 weeks) were used.

### 4.21. Single Cell RNAseq Data

Single cell data count matrices from Adams et al. [55] were downloaded from GEO (accession number GSE136831) and processed using Scanpy [76] (v 1.6). Doublets were identified and removed using Scrublet [77]. Samples were integrated using scVI [78] v0.7. scVI’s latent variables were used for UMAP and for clustering the data using the Leiden method. Epithelial cells were identified as EPCAM+ and isolated from the whole lung set. The epithelial cell types were annotated by identifying the clusters expressing the following marker genes: basal: KRT5, KRT17; aberrant basaloid: COL1A1, KRT17; mucous: MUC5AC; club: SCGB3A2; ciliated: TPPP3; AT1: AGER; AT2: SFTPC, SFTPA1, ionocytes: FOXI1.

### 4.22. Statistical Analyses

Two tailed, unpaired or paired *t*-test with Welch-correction (used only for unequal variances) was used to calculate statistical significance for pairwise comparisons. Except for Figure 5d and Figure 6c, one tailed unpaired *t*-test and two tailed paired *t*-test were applied, respectively. Normal distribution was assumed for all analyzed data. Statistical analyses were carried out using the GraphPad Prism software (version 6.01, San Diego, CA, USA). The following indications of significance were used throughout the manuscript: * *p* < 0.05, ** *p* < 0.01, *** *p* < 0.001. Results are shown as the mean ± standard error of the mean (SEM).

## Figures and Tables

**Figure 1 ijms-23-07749-f001:**
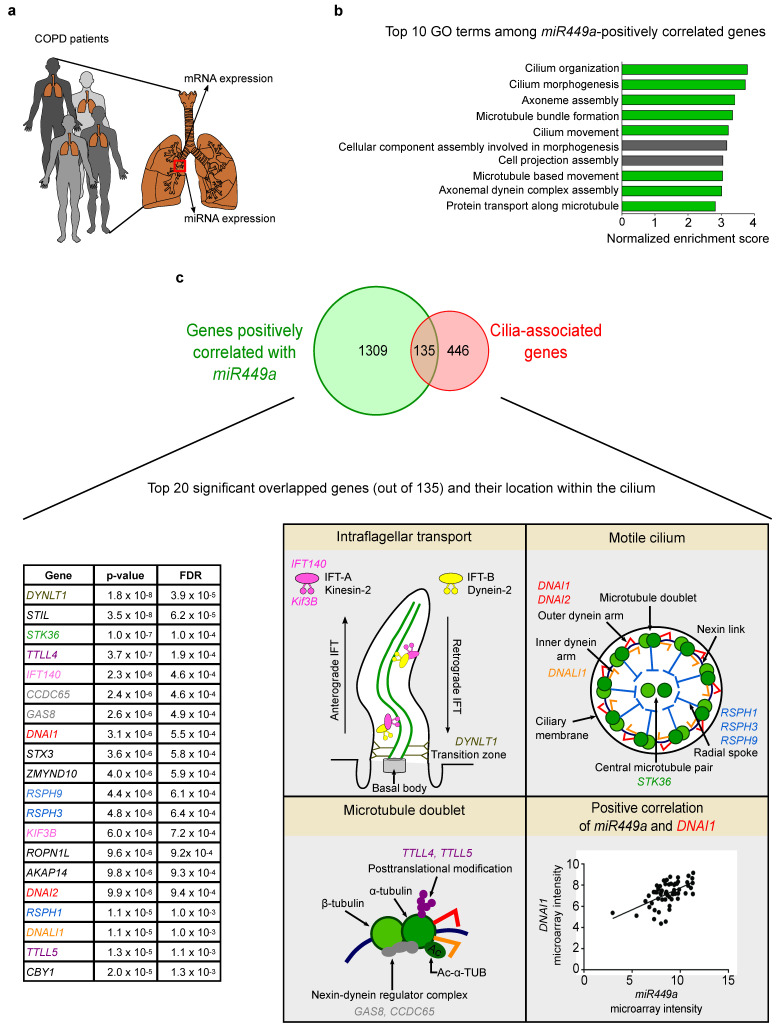
*miR449* expression is positively correlated with genes associated with ciliogenesis in COPD patients. (**a**) Schematic overview of the GLUCOLD study design. Microarray analysis was performed for mRNAs and miRNAs using bronchial biopsies, which were collected from 57 COPD patients diagnosed “with moderate to severe symptoms”. (**b**) Enrichment of the top 10 biological processes among *miR449a*-positively correlated genes. The GSEA was conducted using the Gene Ontology gene set for biological processes and the list of genes ordered by the strength of their correlation to *miR449a*. Green bars mark cilia-associated processes. All displayed gene sets for biological processes are significant with FDR < 0.001. (**c**) Comparison between *miR449a*-positively correlated genes in COPD patients and cilia-associated genes identifies 135 overlapping ciliary genes [26]. Left: table displays the top 20 significantly overlapped ciliary genes (out of 135). Right: illustration of the intraflagellar transport, cross-section of a motile cilium, and of a microtubule doublet highlights the location of some of the top 20 overlapped ciliary genes positively correlated with *miR449a*. Lower right panel: *miR449a* expression is positively correlated with *DNAI1*. Significance was determined by FDR < 0.05.

**Figure 2 ijms-23-07749-f002:**
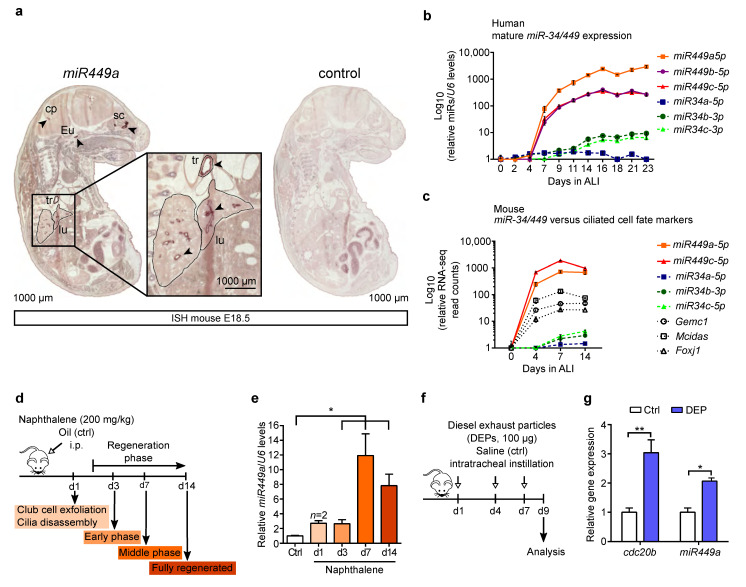
*miR449* is induced upon multiciliated epithelial (re)generation. (**a**) ISH of mouse embryo (E18.5) sections using an *miR449a* probe (**left**) and a secondary antibody control (**right**). Multiciliated organs show positive staining for *miR449a* (arrowheads). Magnification indicates the expression of *miR449a* in trachea and lungs. Cp = choroid plexus, eu = Eustachian tube, sc = supranasal cavity, tr = trachea, and lu = lung. (**b**) Mature *miR34/449* expression in human airway epithelial cells during multiciliated epithelia formation at ALI. Day 0 is used as reference. Data points for *miR449a and miR34a* were previously published [8]. *n* = 3 per time point. (**c**) RNA sequencing read count data [26] for *miR34/449* family members and multiciliated cell fate markers (*Foxj1*, *Gemc1*, and *Mcidas)* during multiciliated differentiation of MTECs at ALI. Day 0 is used as reference. miRNAs: *n* = 2 per time point; mRNAs: *n* = 3 ALIs. (**d**) Schematic representation of naphthalene-induced airway damage and regeneration timeline. Mice were treated once with naphthalene (200 mg/kg, or oil (Ctrl); intraperitoneally (i.p.)). (**e**) Quantification of mature *miR449a* levels in WT lungs at various time points after naphthalene injection. Ctrl levels are used as reference. *n* = 3 (Ctrl), *n* = 2 (d1), and *n* = 5 (d3, d7, d14) mice per group. (**f**) Schematic Illustration of diesel exhaust particles’ (DEPs’) treatment to induce acute lung inflammation in mice. Mice were instilled intratracheally with DEPs (100 µg) or saline (Ctrl) on day 1, 4, and 7, and analyzed at d9. (**g**) Mature *miR449a* and *Cdc20b* levels were quantified in DEP- and saline-treated lungs (Ctrl), and normalized to *U6* or *36b4*, respectively. Ctrl levels are used as reference. *n* = 6 (Ctrl), *n* = 8 (DEP) mice per group. All data are presented as the mean ± SEM with * *p* < 0.05 and ** *p* < 0.01.

**Figure 3 ijms-23-07749-f003:**
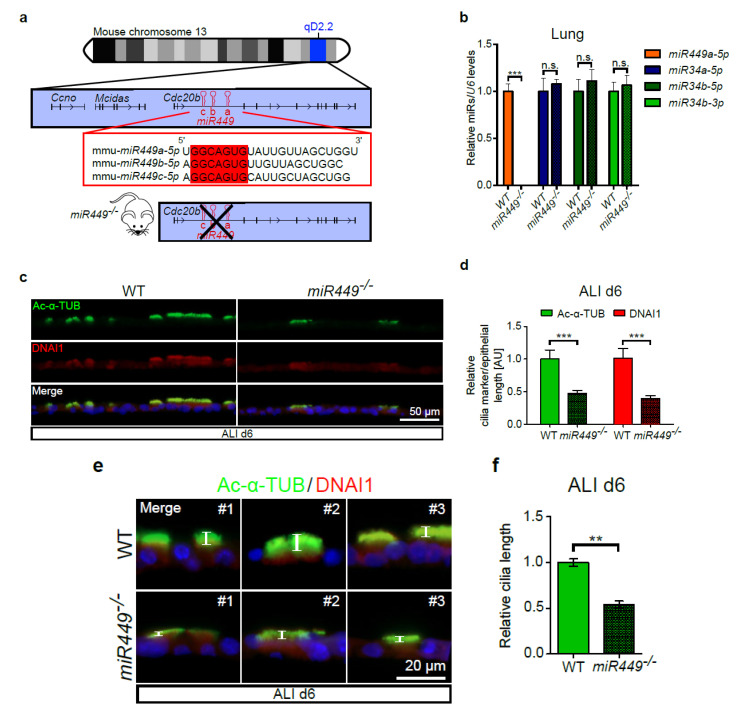
*miR449* depletion impairs ciliation of airway epithelial cultures. (**a**) A scheme of murine chromosome 13, where *Ccno*, *Mcidas*, *Cdc20b*, and *miR449* (plus strand) genes are located. *miR449* cluster is positioned within the second intron of *Cdc20b*. *miR449* cluster member seed sequences are marked in red. Depletion of *miR449a*, *b*, and *c* in *miR449^−/−^* mice occurred without impairing *Cdc20b*. (**b**) Relative expression of mature *miR-34/449* in lungs from WT and *miR449^−/−^* mice. Taqman qPCR, *n* = 5 mice per genotype. (**c**) Immunofluorescence (IF) staining for axonemal cilia markers Ac-α-TUB (green) and DNAI1 (red) in WT and *miR449^−/−^* ALI cultures (d6). Nuclei were stained with DAPI (blue). For more images see Appendix A. (**d**) Quantification of cilia markers IF signal per epithelial length of ALIs shown in **c** relative to WT. *n* = 63 (WT) and *n* = 99 (*miR449^−/−^*) images from 3 WT and 3 *miR449^−/−^* cultures (3 mice per culture). (**e**) Representative higher magnifications from **c** highlighting cilia length (white bars). (**f**) Relative length of cilia measured from d. All data are presented as the mean ± SEM and relative to the WT group with ** *p* < 0.01 and *** *p* < 0.001.

**Figure 4 ijms-23-07749-f004:**
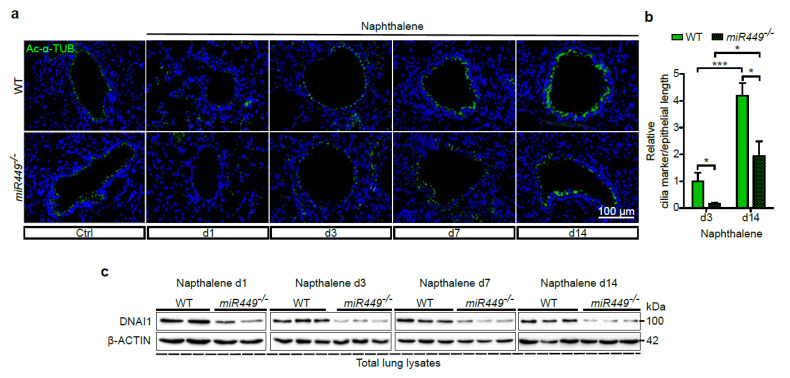
*miR449^−/−^* mice show impaired multiciliated airway epithelial regeneration. (**a**) IF staining for cilia markers Ac-α-TUB (green) and DAPI (blue) of lung sections from vehicle and naphthalene-treated WT and *miR449^−/−^* mice. Cf. Figure 2d illustrates the experimental setting. (**b**) Quantification of Ac-α-TUB IF signal per epithelial length in WT and *miR449^−/−^* lungs at d3 and d14 after naphthalene injury. Levels of d3 WT lungs serve as reference. d3: *n* = 6 mice/genotype; d14: WT *n* = 5 and *miR449^−/−^ n* = 3. (**c**) Motile cilia marker DNAI1 was quantified by immunoblot analysis on samples from experiment in Figure 4a with β-ACTIN used as loading control. (**d**) Treatment scheme of NTHi-induced chronic lung inflammation. WT and *miR449^−/−^* mice were treated with nebulized NTHi for 1 h/d and 5 d/week (w) over a period of 3 months. (**e**) IF for Ac-α-TUB (green) and DAPI (blue) of lung sections from NTHi-treated WT and *miR449^−/−^* mice. (**f**) Immunoblot analysis for motile cilia marker DNAI1 in lung lysates from experiment in Figure 4d; β-ACTIN was used as loading control. (**g**) Expression levels of *Dnah5* analyzed by qPCR and normalized to *36b4* in WT and *miR449^−/−^* lungs. *n* = 4/genotype. (**h**) IF staining for Ac-α-TUB (green), DNAI1 (red), and DAPI (blue) of tracheal sections from 6-month-old WT and *miR449^−/−^* mice. (**i**) Ac-α-TUB IF signal quantification per epithelial length in tracheal sections shown in (**h**). *n* = 3. (**j**) Immunoblot for DNAI1 in lung lysates from 6-month-old WT and *miR449^−/−^* mice; constitutive protein HSC70 was used as a loading control. All data are presented as the mean ± SEM and relative to the WT group with * *p* < 0.05 and *** *p* < 0.001.

**Figure 5 ijms-23-07749-f005:**
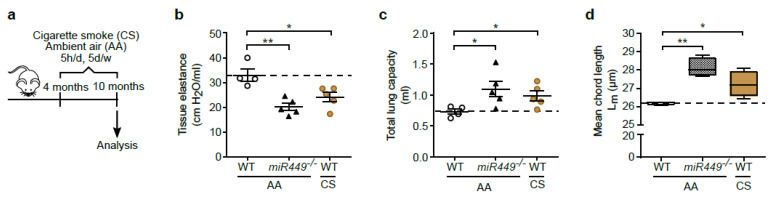
*miR449* deficiency results in a spontaneous COPD phenotype with increased inflammatory response. (**a**) Schematic illustration of the COPD mouse model, smoke-induced. The treatment of WT mice at the age of 4 months took place over a period of 6 months, for 5 h/d and 5 d/w. WT mice were either exposed to CS or maintained at AA, whereas *miR449^−/−^* were exposed, during this period, to AA. A positive control for the development of the COPD phenotype were WT mice exposed to CS. (**b**,**c**) Tissue elastance (**b**) as well as total lung capacity (**c**) were measured to assess pulmonary function using *miR449^−/−^,* CS-exposed WT and WT mice at 10 months of age. *n* = 4–5 mice/genotype. Analysis was conducted using the FlexiVent system. (**d**) By measurements of the mean chord length, emphysema was analyzed in *miR449^−/−^,* CS-exposed WT and WT mice at the age of 10 months. *n* = 4 mice/genotype. (**e**) Counting of macrophages in BALF of *miR449^−/−^,* CS-exposed WT and WT mice at 10 months of age. *n* = 5 mice/genotype. (**f**) Immunoblot analysis of M2 macrophage marker CD206 in lysates of WT and *miR449^−/−^* mice following NTHi treatment. HSC70 was used as a loading control and quantification thereof (**g**), respectively; WT serves as reference. (**h**) Relative mRNA expression of *Mmp9* and *Mmp12* in WT and *miR449^−/−^* mice treated with NTHi, normalized to *36b4.* WT levels serve as a reference. *n* = 4 mice/genotype. (**i**) Immunoblot analysis of TIMP1 protein expression and relative quantification thereof (**j**). HSC70 was used as loading control. All values are presented as the mean ± SEM and relative to the WT group with * *p* < 0.05, ** *p* < 0.01, and *** *p* < 0.001.

**Figure 6 ijms-23-07749-f006:**
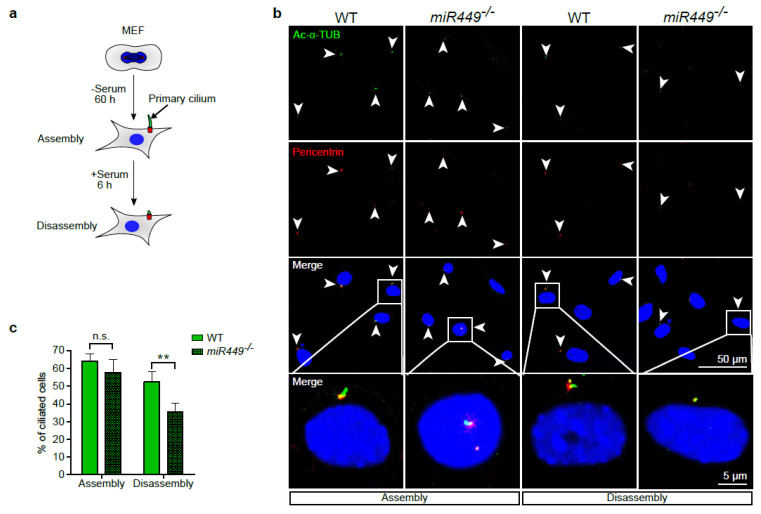
Targeting Aurora kinase A via *miR449* reduces ciliary disassembly. (**a**) Schematic illustration of the experimental procedure to induce assembly and disassembly of primary cilia in MEFs. Serum removal for 60 h triggers primary cilium assembly, whereas the subsequent addition for 6 h leads to ciliary disassembly. (**b**) Visualization of the ciliary axoneme as well as the basal body, respectively, in Ac-α-TUB (green) and pericentrin (red) stained MEFs of WT and *miR449^−/−^* mice. Nuclei were counterstained using DAPI (blue). Primary cilia are indicated by arrowheads. Primary cilia are highlighted in the lowest panel in higher magnification from boxed regions in the panel above. (**c**) Quantification of ciliated cells, respectively, in MEFs of WT and *miR449^−/−^* at points in time of assembly and disassembly. (**d**) Schematic illustration of ciliary disassembly (schematic overview was adapted from [35]). Ca^2+^ influx into the cytoplasm is triggered by growth factor stimulation and results in binding of Ca^2+^ to calmodulin (CaM). CaM subsequently binds to, and thus activates AURKA by autophosphorylation, which, in turn, phosphorylates HDAC6. In the following, HDAC6 removes acetyl groups from α-tubulin, which triggers ciliary disassembly [35,44]. Kv10.1 (*KCNH1*) promotes, via an unknown mechanism, ciliary disassembly [36]. (**e**,**f**) H1299 cells were transfected with pre-miRNAs from the *miR449* family in addition to a natural miRNA (*miR302**) without predicted AURKA binding sites and a synthetic control sequence (Ctrl; NC#2), both as references. Forty-eight hours post-transfection, lysates were harvested and AURKA protein and mRNA expression were analyzed in immunoblots (**e**) and by qPCR ((**f**), *n* = 3). (**g**) Immunoblot analysis of AURKA protein levels in lungs of 6-month-old *miR449^−/−^* and WT mice, and relative quantification thereof. β-ACTIN was used as loading control. *n* = 4 mice per genotype. (**h**) AURKA protein expression in lung lysates isolated from naphthalene-treated WT and *miR449^−/−^* mice at d7. β-ACTIN was used as loading control. All data are presented as the mean ± SEM and relative to the WT group with * *p* < 0.05 and ** *p* < 0.01.

**Figure 7 ijms-23-07749-f007:**
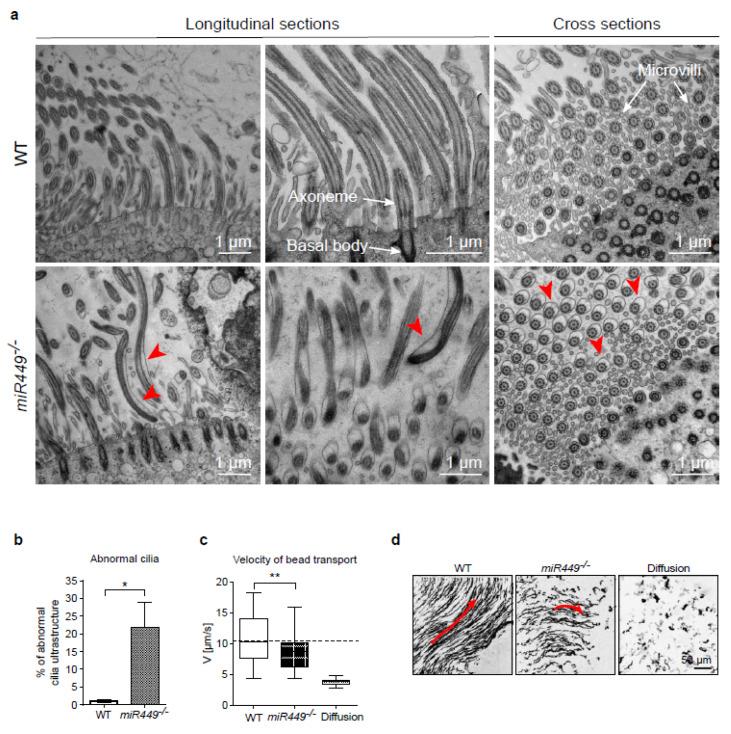
*miR449* deficiency causes cilia ultrastructural defects and reduced mucociliary transport. (**a**) Transmission electron microscopy (TEM) was used to inspect tracheal cilia of WT and *miR449^−/−^* mice. Red arrowheads point to aberrant cilia with membrane detachment. (**b**) Percentage of aberrant cilia observed in TEM with *n* = 3 WT (711 cilia from 168 images) and *n* = 5 *miR449^−/−^* (305 cilia from 184 images). (**c**) Velocity of bead transport in WT and *miR449^−/−^* tracheae. Passive diffusion over dead tracheae of both genotypes serves as control. *n* = 38 WT measurements on 5 mice; *n* = 36 *miR449^−/−^* measurements on 5 mice; *n* = 25 diffusion measurements on 9 mice. Appendix A contain bead transport movies. (**d**) Bead trajectories aggregated from 2000 images over 32 s of video. WT tracheae display longer trajectories along a flow field. Direction of beads is indicated by red arrows. All data are displayed as the mean ± SEM and compared to the WT group with * *p* < 0.05 and ** *p* < 0.01, respectively.

**Figure 8 ijms-23-07749-f008:**
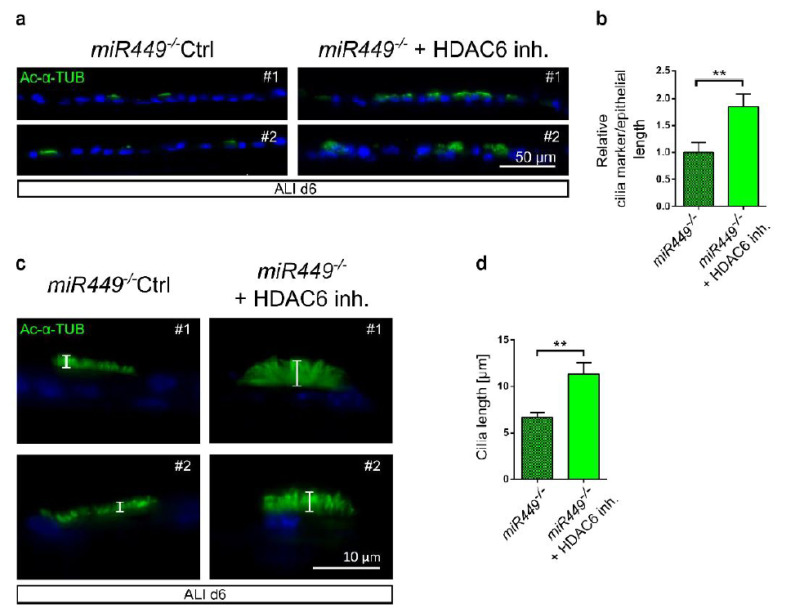
Inhibition of HDAC6 rescues ciliation in *miR449^−/−^* ALI cultures. (**a**) *miR449^−/−^* ALI cultures were treated with DMSO (Ctrl) or tubastatin (100 nM), a HDAC6 inhibitor (inh.), twice daily from d0 to d6 under serum-free conditions. At d6 cultures were fixed and stained for Ac-α-TUB (green) and nuclear DNA (DAPI, blue). (**b**) Quantification of IF signal of cilia marker per epithelial length of ALIs shown in (**a**) relative to *miR449^−/−^* Ctrl. *n* = 19 (*miR449^−/−^* Ctrl) and *n* = 24 (*miR449^−/−^* + HDAC6 inh.) images from 2 *miR449^−/−^* (Ctrl) and 2 *miR449^−/−^* + HDAC6 inh. ALI cultures (3 mice per culture). (**c**) Representative images of cilia length, (**d**) Cilia length is measured from (**a**,**c**). All data are presented as the mean ± SEM and relative to the *miR449^−/−^* Ctrl group with ** *p* < 0.01.

**Figure 9 ijms-23-07749-f009:**
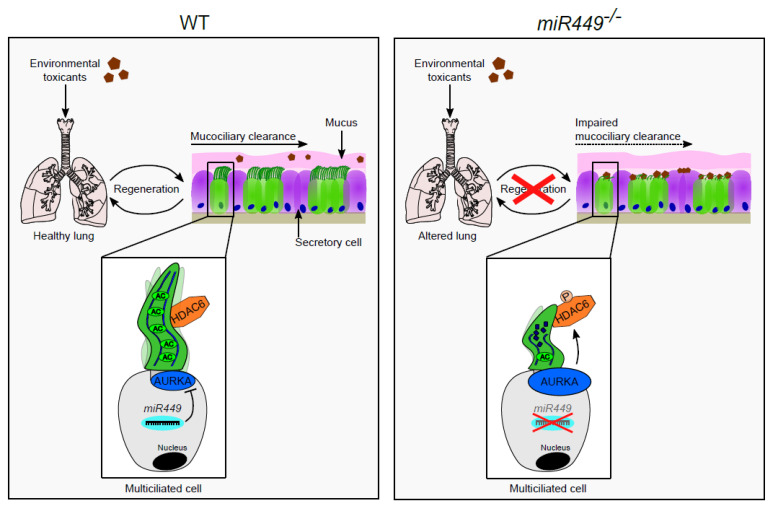
miR449 protects airway regeneration (schematic model). **Left** panel (WT mice): When environmental toxicants (bacteria, cigarette smoke, diesel exhaust particles, naphthalene) enter the lungs, they are normally trapped in mucus and transported out of the airways by synchronized beating of numerous motile cilia, a phenomenon known as mucociliary clearance. Adequate clearance enables healthy airway regeneration, thereby preventing chronic airway inflammation. *miR449* ensures cilia maintenance and healthy clearance by controlling AURKA/HDAC6-mediated ciliary disassembly. **Right** panel (*miR449^−/−^* mice): by promoting AURKA-mediated ciliary disassembly, loss of miR449 affects ciliated epithelium renewal and mucociliary clearance, resulting in stronger inflammatory response to insult as toxicants may not be cleared efficiently, ultimately resulting in the development of emphysematous manifestations of COPD. Legend: AC = acetylated-*alpha*-tubulin, P = phosphorylated HDAC6.

**Figure 10 ijms-23-07749-f010:**
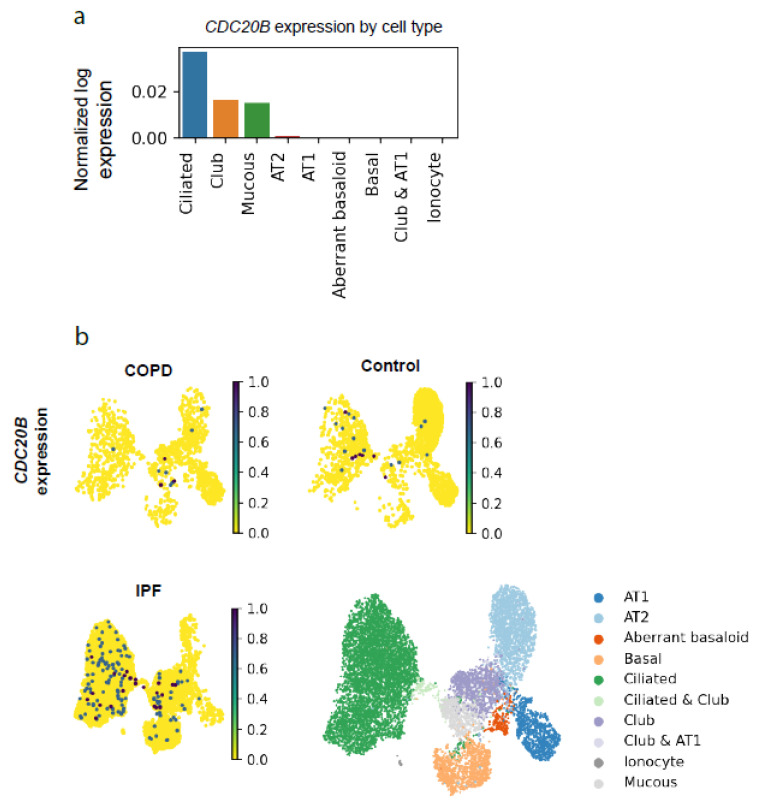
Expression of *CDC20B* and *HDAC6* in public single cell data from [54] was downloaded and reprocessed (**a**) Mean expression of *CDC20B* in epithelial cells from healthy controls (*n* = 16). (**b**) *CDC20B* expression in UMAP dimensionality reduction, comparing between COPD (*n* = 14, cells = 1390), control (*n* = 16, cells = 2666), and IPF (*n* = 32, cells = 6592). Lower right: UMAP plot shows the annotation of cell types in the dataset. (**c**) Dot plot showing the expression of *CDC20B*, *AURKA*, and *HDAC6* in ciliated cells. The normalized log expression is scaled from lowest (0) to highest (1).

## Data Availability

Single cell data count matrices from Adams et al. [55] were downloaded from GEO (accession number GSE136831).

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
