# Peer review of "miR449 Protects Airway Regeneration by Controlling AURKA/HDAC6-Mediated Ciliary Disassembly"

_ijms, 2022, doi:10.3390/ijms23147749_

Round 1

Reviewer 1 Report

The article by Wildung et al. (ijms-1764863) “miR449 protects airway cilia and healthy lung aging” is a complete study that establishes a link between the microRNA miR449, ciliary dysfunction, and Chronic obstructive pulmonary disease (COPD). The authors show that miR449 is involved in mucociliary regeneration, protects airway cilia, and in the last term healthy lung aging.

miR449 was already described in the literature as dysregulated in ciliated cell differentiation pathways and is essential in regulating airway ciliated cells by targeting NOTCH1 (reference 9 of this manuscript), being involved in Asthma and Respiratory Infections. The present work delves into previous findings related to this topic, including some of this manuscript authors, therefore, with this article we further understand the role of RNA-mediated regulation in lung diseases. They offer a well-made discussion with some figures, one of them especially illustrative (Figure 9).

I have some general comments that could help to get an improved version:

Major comments:

-     I have an issue with the title of the manuscript (miR449 protects airway cilia and healthy lung aging). It looks too general, perhaps more appropriate for a Review manuscript, and addresses the conclusions of this study in an indefinite way.

-     The authors made a vague description of the function exerted by miRNAs on their target mRNAs (Lines 70-72). I suggest explaining this point better:

o   What is the role of ribonucleases during the mRNA degradation/translational repression?

o   How about the hybridization between miRNAs-mRNAs? is it perfect or incomplete complementarity?

o   Could this union to the 3’UTR of the target cause in any case its destabilization?

o   It has been described that miRNAs can switch between activating and repressing target gene expression. They also can bind to 5′ UTR and coding regions, and positively-regulate gene transcription by targeting elements in the promoter region.

-     Figure 1: were there healthy subjects to analyze bronchial biopsies as a comparative control? Its absence must be suitably justified. Regarding this, is it possible that section 2.1. “Cilia-related genes positively correlate with miR449 expresssion in COPD patients” could have more sense being explained in a different rather than first place?

Did not the authors include differentiation in the results obtained between patients “with moderate to severe symptoms” in Figure 1?

-     The technique or antibodies used in the western blotting could be optimized to avoid the bands' issue in Figure 5, i), Figure 6, g), h).

Minor comments:

·       Line 33: please, describe the meaning of “-/- mice” (both alleles of the gene have been knocked out?) the first time that is mentioned.

·       Lines 67-69: could the authors further introduce the post-transcriptional regulation prior to the miRNAs mention?

·       Line 107: Aurora kinase A should be further introduced.

·       Line 134: please, mention the GLUCOLD study in the main text where Figure 1 is analyzed, indicating that it will be described in the material and methods section.

·       Line 133/Figure 1: Could the resolution be improved?

·       Line 158: why 23 days?

·       Lines 207-210: not sure that I understood the relevance of testis analysis for this essay.

·       Line 300: please, avoid the use of “data not shown” in the main results section. If such results are not directly relevant, discuss them in the appropriate section.

·       Line 365/Figure 6, b): could the resolution be improved?

·       Line 542/Figure 10: please, improve the resolution of b) section.

Author Response

Thank you for your valuable comments, please find the attachment

Reviewer 2 Report

The manuscript by Wildung et al reports the involvement of the miR449 in the ciliogenesis and its connection with cilia maintenance in the in vitro polarized epithelium culture model and the airways of miR449-/- mice. Authors also identified the aurora kinase A and its effector target HDAC6 in the miR449-regulated-ciliary homeostasis and epithelial regeneration. The study was well-designed, data analyses were reliable and the results and discussion sections were well written. Reviewer recommends its publication in IJMS. 

Minor suggestions:

1)    The figure 1a sounds unnecessary as clearly described in the text and Stable S1;

2)    Check page 6, line 207, “whereas miR43a nor miR34b, c are not”, nor or or?

3)     Check Figure 3 legend, page 7 lone 231, “f) Relative length of cilia measured from d”. Should be from c? 

Author Response

(The authors gave the same response as above.)
